# Oral Ingestion of Bacterially Expressed dsRNA Can Silence Genes and Cause Mortality in a Highly Invasive, Tree-Killing Pest, the Emerald Ash Borer

**DOI:** 10.3390/insects11070440

**Published:** 2020-07-14

**Authors:** Ramya Shanivarsanthe Leelesh, Lynne K. Rieske

**Affiliations:** 1Department of Entomology, University of Kentucky, Lexington, KY 40546-0091, USA; ramya.sl1989@gmail.com; 2School of Life Sciences, University of Bedfordshire, Luton LU13JU, UK

**Keywords:** RNA interference, RNAi, *Agrilus planipennis*, EAB, RNAi-based biopesticide, forest pest management

## Abstract

RNA interference (RNAi) is a naturally occurring process inhibiting gene expression, and recent advances in our understanding of the mechanism have allowed its development as a tool against insect pests. A major challenge for deployment in the field is the development of convenient and efficient methods for production of double stranded RNA (dsRNA). We assessed the potential for deploying bacterially produced dsRNA as a bio-pesticide against an invasive forest pest, the emerald ash borer (EAB). EAB feeds on the cambial tissue of ash trees (*Fraxinus* spp.), causing rapid death. EAB has killed millions of trees in North America since its discovery in 2002, prompting the need for innovative management strategies. In our study, bacterial expression and synthesis of dsRNA were performed with *E. coli* strain HT115 using the L4440 expression vector. EAB-specific dsRNAs (*shi* and *hsp*) over-expressed in *E. coli* were toxic to neonate EAB after oral administration, successfully triggering gene silencing and subsequent mortality; however, a non-specific dsRNA control was not included. Our results suggest that ingestion of transformed *E. coli* expressing dsRNAs can induce an RNAi response in EAB. To our knowledge, this is the first example of an effective RNAi response induced by feeding dsRNA-expressing bacteria in a forest pest.

## 1. Introduction

RNA interference (RNAi) regulates gene expression at the post-transcriptional level by degrading specific messenger RNAs (mRNA), thus blocking translational efficiency [1]. RNAi using exogenous dsRNA is emerging as a novel means of pest suppression [2]. After introduction into cells, dsRNA is recognized by the RNase III enzyme dicer and processed into small interfering RNAs (siRNAs). These siRNAs then bind to the Argonaute protein and form an RNA-induced silencing complex (RISC), and the RISC complex binds to the complementary mRNA molecule, thus blocking gene expression [3].

Coleopteran insects are known to exhibit robust RNAi responses [2,4,5]. RNAi efficiency varies between insect species, insect life stages, target genes, and modes of dsRNA delivery [6]; dsRNA can be delivered in several ways, including by injection, orally, and through absorption [7]. While RNAi is emerging as an attractive option for insect pest control, convenient and efficient methods to produce and deliver dsRNA to target insects is challenging.

The emerald ash borer (EAB), *Agrilus planipennis* Fairmaire, is an exotic beetle that was accidentally introduced from China into North America in the mid- to late 1990s [8]. Adult beetles feed on ash, *Fraxinus* spp., and foliage and cause little damage, but larvae feed on cambial tissue beneath the bark, disrupting water and nutrient flow, and causing rapid tree death [9]. Ash species native to North America have very little resistance to the emerald ash borer [10]. EAB has killed millions of ash trees in North America since its introduction [11], and the invasion continues. Chemical suppression can be effective, but is expensive and unsustainable over large areas [12,13]. Classical biological control has been implemented, but it is slow-acting and expensive [14]. Thus, the EAB invasion in North America warrants explorations into innovative approaches for management [15].

In this study we evaluated the insecticidal potential of dsRNA-expressing bacteria delivered orally to neonate EAB larvae. dsRNA expressed in bacteria could provide dual benefits in terms of inexpensive production and efficient delivery.

## 2. Materials and Methods

### 2.1. Insect Rearing

EAB eggs were obtained from the United States Department of Agriculture, Animal and Plant Health Inspection Service, Plant Protection Quarantine (USDA APHIS PPQ) EAB Biocontrol Facility (Brighton, MI, USA). Immediately upon receipt the eggs were placed in Petri dishes (60 × 90 mm) with moistened filter paper and maintained at 23 °C and 75% relative humidity in a growth chamber. Newly hatched unfed larvae were used in bioassays.

### 2.2. Target Gene Selection, Total RNA Extraction, cDNA Synthesis, PCR Amplification, and Construction of Recombinant L4440 Vector

To assess the insecticidal activity of bacterially-expressed dsRNA, candidate genes shibire (*shi*) and heat shock protein-70kDA (*hsp*) were chosen due to their effectiveness in RNAi-induced EAB mortality by in vitro produced dsRNA [15]. Total RNA was extracted from EAB larvae using Trizol reagent (Invitrogen, Carlsbad, CA, USA) according to the manufacturer’s instructions; RNA concentration and purity were evaluated using a Nano Drop 1000 (Thermo Fisher, USA). cDNA was synthesized from 1 µg of total RNA using a M-MLV reverse transcriptase kit (Thermo Fisher, USA). Target sequences of *shi* and *hsp* were amplified using gene-specific primers with restriction enzymes (Table 1). PCR conditions were as follows: 94 °C for 5 min, followed by 35 cycles of initial denaturation at 94 °C for 30 s, annealing at 60 °C for 30 s and extension at 67 °C for 1 min, and finishing with extension at 67 °C for 8 min.

The L4440 plasmid (Addgene plasmid 1654) comprising two T7 promoters in an inverted position flanking multiple cloning sites was used to clone target genes. Restriction (Xba I and XmaI) digested amplicons were ligated into the Xba I and XmaI digested L4440 vector, respectively. The recombinant vectors were validated by colony PCR [16] and restriction digestion (Xba I and XmaI).

### 2.3. Bacterial Transformation and Expression of dsRNA

The RNase III-deficient *E. coli* strain HT115 (DE3), obtained through the CGC at the University of Minnesota, was grown in Luria broth (LB) medium with ampicillin (100 µg/mL) and tetracycline (10 µg/mL). The recombinant L4440 vector was transformed into *E. coli* HT115 (DE3) competent cells. Single bacterial colonies were cultured in LB broth maintained on a shaker incubator at 37 °C (225 rpm) overnight. Cultured broth was then transferred to 50 mL fresh broth medium containing 100 µg/mL ampicillin and cultured at 37 °C until colony growth reached the late exponential phase, with OD_600_ = 0.4–0.6. Expression of T7 RNA polymerase was induced by adding a final concentration of 0.5 mM of isopropyl-β-D-1-thiogalactopyranoside (IPTG). Bacteria with dsSHI and dsHSP were then incubated at 37 °C, 30 °C, and 25 °C for up to 4 h to evaluate dsRNA expression. Based on this optimization, further experiments were conducted at 37 °C for dsSHI and 30 °C for dsHSP. Total bacterial RNA was extracted with Trizol reagent and the presence of synthesized dsRNA was confirmed by electrophoresis using a 1% agarose gel.

### 2.4. Biological Activity of Recombinant Bacteria Expressing dsRNA

Laboratory-reared EAB eggs were placed in petri dishes with moistened filter paper and maintained at 24 ± 1 °C in an incubator (Figure 1). Newly hatched neonates were used in all bioassays. To determine the biological activity of the recombinant bacteria expressing dsRNA, neonate EAB larvae were fed using a modified droplet feeding bioassay [5], where 1 ml of bacterial culture was centrifuged at 3000 rpm for 15 min and the pellet was dissolved in 100 μL of 1% sucrose solution with green tracking dye (Kroger, Co., USA). For each assay, 3μL of bacterial suspension were fed to individual neonate larvae using the droplet assay. Cellular density of the bacterial culture was determined by considering that an optical density of 1 at 600 nm corresponds to 10^8^ bacterial cells/mL [17]. As the control, HT115 (DE3) bacteria were used as a treatment [18]. Neonate EAB larvae were fed dsRNA-expressing bacteria for five consecutive days. On day 6, larvae were fed with 1% sucrose lacking dsRNAs for two days. Assays were maintained in an incubator at 26 ± 1 °C, under a 14:10 (L:D) photoperiod. Mortality was measured on day 5 (the last day of dsRNA feeding) and on day 7 (the final day of bioassays). Each treatment was replicated three times, and for each replication, 10–15 larvae were used. Mortality (%) was calculated and the mean values of the experimental replicates were analyzed using a one-way ANOVA, with Tukey’s post-hoc t-test to evaluate differences.

### 2.5. Molecular Validation of Gene Silencing

Following ingestion of dsRNA-expressing bacteria, total RNA was isolated from 5–6 EAB larvae at two time intervals (72 h and 120 h) using Trizol reagent. Total RNA was treated with DNase I to degrade any genomic contamination. cDNA was synthesized using a M-MLV Reverse Transcriptase Kit (Thermo Fisher, USA), and was used as a template for gene expression studies. The expression analysis of the target gene was performed using SYBR™ Green Master Mix (Applied Biosystems, USA). qPCR reactions were performed using StepOnePlus Real-Time PCR system (Life Technologies, USA). All reactions were carried out in triplicate with a final volume of 10 μL. A melting curve was generated at the end of each reaction to confirm single product (target) amplification. In order to eliminate undesirable amplification from input recombinant plasmids and/or dsRNAs, primers for qPCR were designed to detect target mRNAs by amplifying only sequences that lay outside of the insert interfering sequences. The TEF1α gene (Table 1) was used as the reference gene ([19]; Appendix A), and the 2^−ΔΔCt^ method [20] was used to calculate expressions of the target gene relative to the control. A two-tailed t-test was used for statistical analysis to compare the means of a single variable.

## 3. Results

### 3.1. Bacterial Transformation and Expression of dsRNA

Bacteria were prepared with the recombinant vector containing fragments of the *shi* and *hsp* genes (Figure 2a). Using colony PCR, we confirmed that 100% of the recombinant bacterial colonies tested contained the insert: dsSHI (483 bp) and dsHSP (468 bp) (Figure 2b), and the IPTG-induced bacteria expressed dsRNA specific to EAB (*shi* (483 bp) and *hsp* (468 bp)). Expression of dsSHI was at 37 °C for 4 h and dsHSP was at 30 °C for 4 h (Figure 2c,d). The two genes were successfully synthesized in the bacteria.

### 3.2. Biological Activity of dsRNA Expressing E. Coli Against EAB

Ingestion of dsRNA-expressing bacteria targeting *shi* and *hsp* caused 69.44% and 46.66% mortality, respectively, of neonate larvae at day 7 (Figure 3 and Figure 4). Larvae ingesting dsSHI and dsHSP experienced greater mortality than control larvae ingesting bacteria that lacked the dsRNA, and dsSHI-ingested larvae appeared to grow more slowly based on larval length (Figure 5).

### 3.3. Molecular Validation of Gene Silencing

Our qPCR analysis showed that bacteria containing dsSHI resulted in a 24.92% reduction in gene expression at 72 h; expression did not differ from controls. However, at 120 h post-exposure there was a 74.14% reduction in the transcript level, which differed significantly from controls (untransformed bacteria) (Figure 6a). Silencing *hsp* caused 48.67% and 96.94% reductions in the transcript level relative to controls at 72 h and 120 h, respectively (Figure 6b), following exposure to bacterially-expressed dsRNA.

## 4. Discussion

This is the first time an effective RNAi response in the tree-killing EAB using oral ingestion of dsRNA-expressing bacteria has been demonstrated. We transformed HT115 *E. coli* to express dsSHI and dsHSP specific to EAB, which caused gene knockdown and showed biocidal activity that resulted in significant mortality of neonate EAB larvae. Use of bacterially-expressed dsRNA to trigger RNAi was first demonstrated experimentally in *Caenorhabditis elegans* [21], and bacterially-expressed dsRNA has subsequently been used against numerous insect pests. In the coleopteran *Leptinotarsa decemlineata*, ingestion of bacterially-expressed dsRNA led to effective suppression of five target genes, causing decreases in body weight and significant mortality of treated beetles [22]. In our work the engineered *E. coli* strain HT115 (DE3), lacking dsRNA-specific RNase III produced EAB-specific dsRNAs and effectively triggered the RNAi pathway upon ingestion by EAB larvae. These features make HT115 (DE3) a promising strain for preparing dsRNA in vivo, providing a less costly and potentially more efficient alternative to in vitro synthesis of dsRNA. However, bacterial dsRNA production can have limitations; recombinant bacterial production of dsRNA is reportedly less effective in causing mortality in *Spodoptera exigua* (Order: Lepidoptera) than is in vitro synthesized dsRNA [23], perhaps due to a lower expression of target gene(s) in bacteria. These direct comparisons have yet to be made experimentally in EAB.

Selection of target gene(s) and target regions within gene(s) is crucial for successful gene silencing. Second-generation sequencing can provide information on target gene selection and screening [4,24], with the goal of selecting genes and/or target regions within genes with increasing RNAi efficiency in the target pest, while having no measurable off-target effects. Here we used two target genes, shibire (*shi*), and heat shock protein-70kDa protein (*hsp*), which play essential biological functions and are efficacious in EAB [15], and demonstrate their potential for use in bacterially-expressed RNAi-based EAB management. The heat shock-70kDA protein gene (*hsp*) functions in protein folding and protects cells from stress [25], while the shibire gene (*shi*) is involved in production of microtubule bundles, endocytosis and other vesicular trafficking processes [26]. The loss in function of either of these target genes in EAB neonates ingesting transformed bacteria causes significant larval mortality (*shi*: ~69% and *hsp*: ~46%) as well as an apparent suppression of larval growth (*shi*). Ingestion of bacteria producing dsRNAs, specifically double stranded integrin (dsINT), has also been shown to reduce growth of the lepidoptera, *S. exigua* [23].

We have demonstrated that, when ingested, bacteria transformed to produce EAB-specific dsRNA can silence target genes and kill neonate EAB, which creates additional potential for its use as a biopesticide. Recombinant bacteria with EAB-specific dsRNA could be sprayed on foliage to be ingested by feeding beetles or on ash stems to be ingested by newly hatched neonates. Naked dsRNAs applied topically or through root drenching can be assimilated into and moved through ash plant tissues [27], suggesting crude extract of bacterially-expressed dsRNA may also be able to be translocated through the plant via soil drench, trunk sprays or injection [28]. Recombinant bacteria producing EAB-specific dsRNAs also make the genetic transformation of ash trees with dsRNA-expressing constructs a more plausible goal [29].

## 5. Conclusions

We showed that EAB fed with dsRNA-expressing bacteria results in downregulation of selected genes, demonstrating the potential for application of bacterially-expressed dsRNA for controlling EAB; however, in the current experiment a non-specific dsRNA control was not included. Although optimization of bacterial dsRNA production and expression is needed, our observations suggest that RNA interference mediated by bacterial dsRNA could be a convenient and cost-effective approach for managing this invasive pest.

The specificity of these EAB-specific dsRNAs is an essential step towards moving this approach to the deployment phase; these evaluations are under way. Additionally, management of resistance in the pest population is essential. Western corn root worm, *Diabrotica virgifera*, has shown field-level resistance to DvSnf7 dsRNA [30], necessitating development and selection of new targets. This process is relatively simple, however, and involves screening for and switching to other appropriate dsRNAs, thereby managing for the potential development of resistance. dsRNAs can be designed for a different region in the same target gene or new genes much more quickly and efficiently than developing a more expensive chemical insecticide [22]. There are clearly knowledge gaps that must be addressed before this technology can be deployed in the EAB–ash system. Potential off-target effects in EAB, mutation of RNAi core machinery genes, mutation of target genes, and enhanced dsRNA degradation, not to mention potential effects on non-target organisms, must be more fully understood before deployment of bacterially-expressed dsRNA in EAB management can become practical.

RNAi is an emerging pest management tool with tremendous potential to protect plants against insect pests. Its application continues to expand into crop and vegetable production [2,6,24,31,32], and horticultural [32,33] and forest systems [5,34,35], and there are numerous native and non-native tree pests that might be appropriate candidates. If RNAi technology can be deployed aggressively along the invasion front to prevent widespread and catastrophic tree losses, this could reduce management costs, freeing up resources for other aspects of integrated forest management [34,35,36,37,38]. In some situations, EAB can be managed with chemical insecticides [13,39] and biological control has shown considerable potential [12,13,14]. However, the need for a rapid-acting, effective, and environmentally friendly approach, such as RNAi, remains high.

## Figures and Tables

**Figure 1 insects-11-00440-f001:**
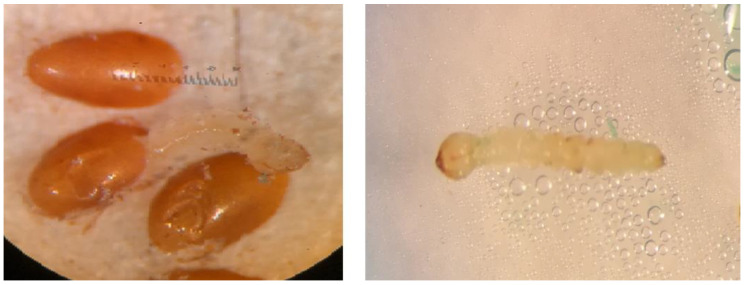
Emerald ash borer egg hatch at 24 ± 1 °C and neonate larva.

**Figure 2 insects-11-00440-f002:**
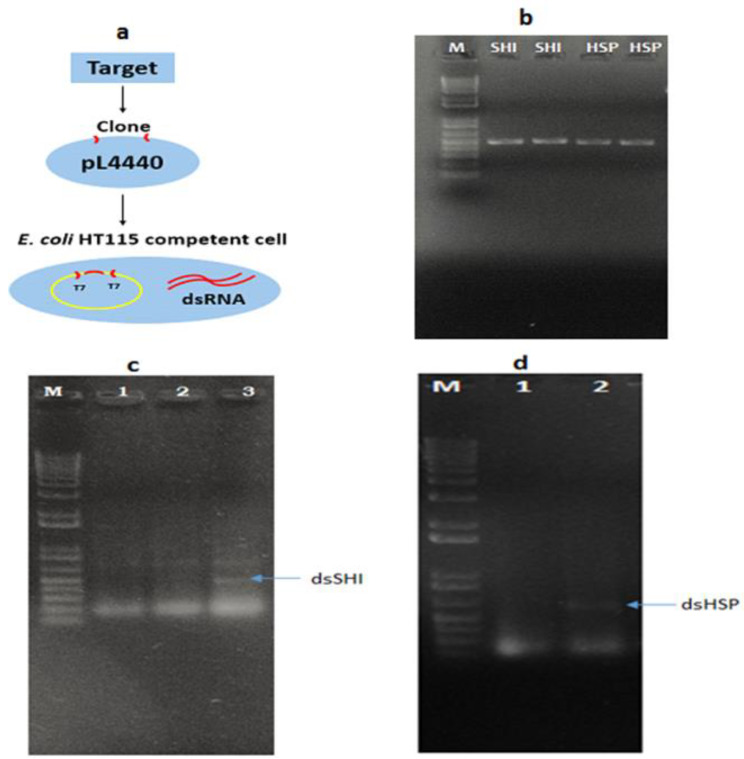
Construction of recombinant *E. coli* expressing emerald ash borer (EAB)-specific dsRNAs, showing (**a**) a schematic diagram of the recombinant plasmid for the expression and production of dsRNA, (**b**) colony PCR confirmation of recombinant bacteria (lane M: 1Kb marker, lane SHI: *shi* gene amplified from two individual bacterial colonies), lane HSP: heat shock protein (*hsp*) gene amplified from two individual bacterial colonies, (**c**) biosynthesis of dsRNA corresponding to partial sequence of the shibire (*shi*) gene in the RNAse III deficient bacterial strain (lane M: 1Kb marker, lanes 1 and 2: uninduced dsSHI, lane 3: dsSHI induced by IPTG), and (**d**) biosynthesis of dsRNA corresponding to partial sequence of the *hsp* gene in the RNAse III deficient bacterial strain (lane M: 1Kb marker, lane 1: uninduced dsHSP, lane 2: dsHSP induced by IPTG).

**Figure 3 insects-11-00440-f003:**
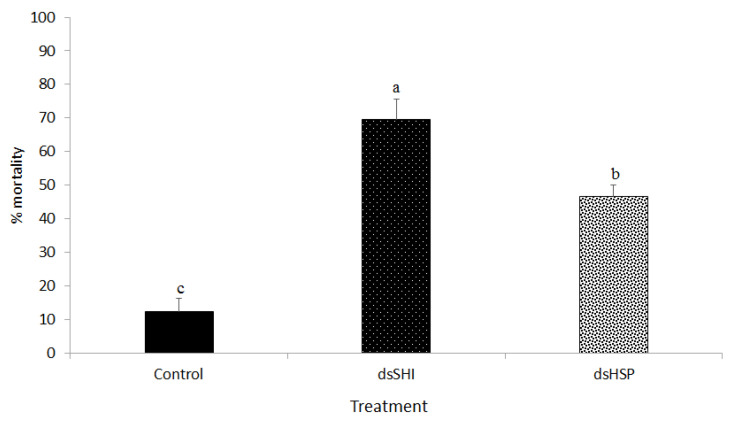
Insecticidal activity of bacterial dsRNA specific to *shi* and *hsp* in EAB larvae is demonstrated by mortality of neonate EAB larvae (mean ± Standard Error) following ingestion of dsRNA-expressing bacteria relative to those ingesting HT115 (control), which contained no dsRNA (*N* = 3). One-way ANOVA, with a post-hoc t-test (Tukey) was used to evaluate differences at *p* < 0.05.

**Figure 4 insects-11-00440-f004:**
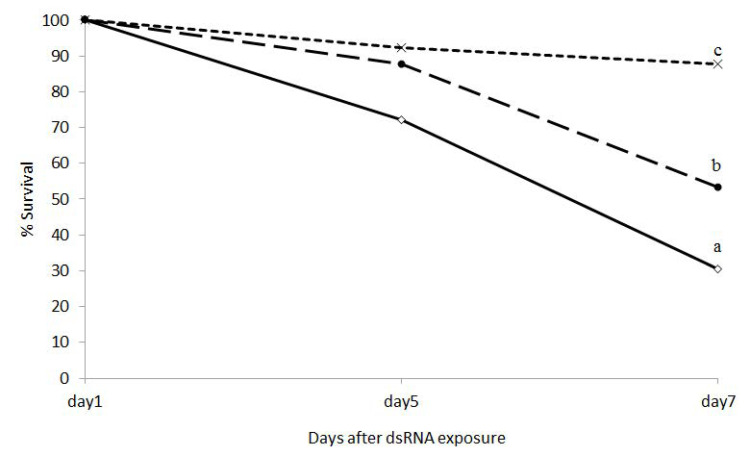
Effect of dsRNAs specific to shi and hsp on EAB neonate larval survival (%) 7 d after feeding on dsRNA-expressing bacteria (*N* = 3). Observations were taken on day 1, day 5, and day 7. One-way ANOVA, with a post-hoc t-test (Tukey) was used to evaluate differences at *p* < 0.05.

**Figure 5 insects-11-00440-f005:**
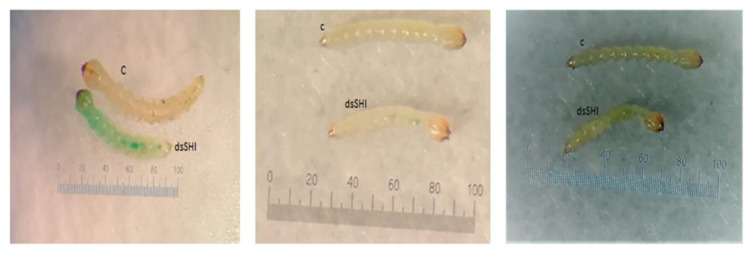
Neonate EAB larvae showing suppressed growth of those fed bacteria expressing dsSHI relative to control larvae (C—control) fed bacteria containing no dsRNA.

**Figure 6 insects-11-00440-f006:**
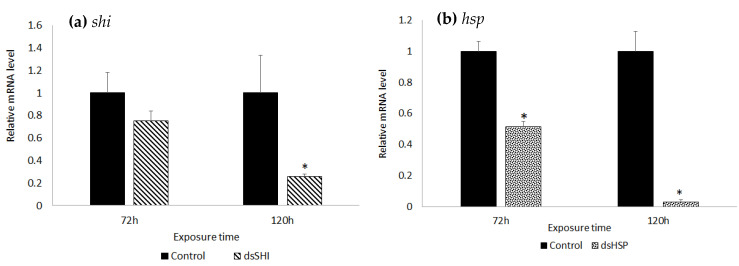
Quantitative RT-PCR analysis of transcript levels after RNAi-mediated repression of gene expression in EAB. Relative expression (mean ± SE) of (**a**) *shi* and (**b**) *hsp* genes in neonate EAB larvae 72 h and 120 h after feeding on dsRNA-expressing bacteria (*N* = 3). Asterisks (*) indicate a significant difference in gene expression within each time interval (t-test, two-tailed *p* < 0.05). Results are expressed as the relative expression of the target gene in treated samples relative to the control.

**Table 1 insects-11-00440-t001:** Primer sequences for dsRNA synthesis and qPCR of the target genes *shi* and *hsp*, and for qPCR of the reference gene TEF-1α; bold italics indicate sequences of restriction enzymes.

Gene	Primer	Primer Sequence (5′–3′)
*hsp*—heat shock protein (70kDA)	F-dsRNA-HSP	***CTAGTCTAGA***GTTACGAGCCAGGGTGAAAA
	R-dsRNA-HSP	***TCCCCCCGGG***CCTTTTGAACGGCACGGTTAT
	F-qRNA-HSP	GACAAAGGAACGGGAAACAA
	R-qRNA-HSP	TCTCGGCATCCCTTATCATC
*shi*—shibire	F-dsRNA-SHI	***CTAGTCTAGA***TGGCACATTTGTATGCCAGT
	R-dsRNA-SHI	***TCCCCCCGGG***CTTGTTGCATTTGCTGAGGA
	F-qRNA-SHI	GGGATCTGCCCAAATTAACA
	R-qRNA-SHI	CCCGTCTGAGTTCTTTCTCG
TEF-1α-Translation elongation factor 1 alpha	F-qRNA-TEF	CATTGAAACCTACGTTGTCGC
	R-qRNA-TEF	ACTGGAGTGCTTAAACCTGG

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
