# Peer review of "Oral Ingestion of Bacterially Expressed dsRNA Can Silence Genes and Cause Mortality in a Highly Invasive, Tree-Killing Pest, the Emerald Ash Borer"

_insects, 2020, doi:10.3390/insects11070440_

Round 1

Reviewer 1 Report

The manuscript by Leelesh and Rieske examines the efficacy of using bacteria that produce dsRNAs to control emerald ash borers (EAB). The authors tested two different dsRNAs expressed in E. coli HT115 cells, which they fed to neonate larvae.  Target gene-knockdown was validated using qRT-PCR and mortality of larvae was significantly greater than that of the negative controls after a 7-day period, using either dsRNA.  The authors contend that this research will lead to an inexpensive method of controlling this serious forestry pest. Although their findings are significant, and this insect is indeed a very serious pest, there are some questions and comments that I’d like to see addressed before the paper is published.

  1. A description of the rearing conditions of the insects is missing. How was a colony maintained?
  2. For qRT-PCR, was a DNase step used or were the primers designed to span introns (i.e. how did they know they were avoiding genomic contamination?)
  3. Why not use a negative control bacterium that expressed GFP (a non-specific dsRNA)? A GFP dsRNA was used for the reference gene selection, so it was readily available, and a bacterium lacking a EAB-specific dsRNA would prove that the mortality was directly related to reduction of the target transcripts (and not to a non-specific response to dsRNA).
  4. The selection of the reference gene TEF should be described in the methods. While this gene scored well using the 4 algorithms, the data in the Supplemental data section do not provide details on the number of replicates, and they do not also show the Cq values for the two genes of interest (shi and hsp). The latter would have been informative to see how closely the target genes’ transcript levels matched those of the references. Why was only one reference gene selected, rather than use the geometric mean of more than one? This latter option is becoming the standard approach for qRT-PCR.
  5. Re: Figure 2 c – there are many bands in the dsRNA lane 3 – what are the other bands and can they be ignored because those same bands are in lanes 1 and 2, which might be non-specific bands from bacteria that have failed to produce dsRNA? (please explain those two lanes).
  6. How were the E. coli culturing conditions optimized? Is there is certain time point after culturing to add the IPTG, and is there a certain time point to harvest bacteria to ensure maximal dsRNA production?
  7. QRT-PCR could be used to quantify dsRNA production, to provide a clear idea of the dose of dsRNA administered to the insects. As it stands, it is unclear what dose of dsRNA the insects received. Could the differences in mortality with the two dsRNAs (shi and hsp) be due to different dsRNA doses?
  8. Figure 3 is unnecessary – the same information is displayed in Figure 4.
  9. Why did the mortality assay end at day 7? It would have been ideal to see the number of treated larvae reaching adulthood, and see long-term effects.
  10. Please add the sample size/#replicates to the caption in Figure 4 – it easier than looking for it in the text.
  11. For Figure 5, three pictures do not really provide any quantitative evidence - why not have one picture and then a graph showing some statistically significant data?
  12. The authors suggest in their concluding remarks that resistance to dsRNAs can be easily overcome by selecting new targets. Some recent studies of dsRNA resistance in the corn rootworm have been published, so please revise this claim, based on those findings.
  13. The authors suggested that root drenching could be used to apply dsRNA to the trees, but will roots readily take up bacteria?

Author Response

Response to Reviewers for insects-832830

Author responses to reviewer comments below in blue type; author changes in manuscript revision is in red type.

Reviewer 1

The manuscript by Leelesh and Rieske examines the efficacy of using bacteria that produce dsRNAs to control emerald ash borers (EAB). The authors tested two different dsRNAs expressed in E. coli HT115 cells, which they fed to neonate larvae.  Target gene-knockdown was validated using qRT-PCR and mortality of larvae was significantly greater than that of the negative controls after a 7-day period, using either dsRNA.  The authors contend that this research will lead to an inexpensive method of controlling this serious forestry pest. Although their findings are significant, and this insect is indeed a very serious pest, there are some questions and comments that I’d like to see addressed before the paper is published.

  1. A description of the rearing conditions of the insects is missing. How was a colony maintained?

This has been incorporated. Rev. lines 54-57 now reads:

Laboratory reared EAB eggs were placed in Petri dishes (60 X 90mm) with moistened filter paper and maintained at 23°C and 75% relative humidity in a growth chamber. Newly hatched unfed larvae were used in bioassays.

  1. For qRT-PCR, was a DNase step used or were the primers designed to span introns (i.e. how did they know they were avoiding genomic contamination?)

The total RNA was treated with DNase I to degrade any genomic contamination. This has been added. (rev. line 115-116)

  1. Why not use a negative control bacterium that expressed GFP (a non-specific dsRNA)? A GFP dsRNA was used for the reference gene selection, so it was readily available, and a bacterium lacking a EAB-specific dsRNA would prove that the mortality was directly related to reduction of the target transcripts (and not to a non-specific response to dsRNA).

The control used in our study proves that the mortality observed is due to the dsRNA expressed by our manipulated bacteria, and not due to the bacteria which was fed to the neonates. This approach is well substantiated in the literature using HT115 cells as control, including:

  • Ganbaatar, O., Cao, B., Zhang, Y. et al.Knockdown of Mythimna separata chitinase genes via bacterial expression and oral delivery of RNAi effectors. BMC Biotechnol 17, 9 (2017). https://doi.org/10.1186/s12896-017-0328-7

  • Vatanparast M, Kim Y. Optimization of recombinant bacteria expressing dsRNA to enhance insecticidal activity against a lepidopteran insect, Spodoptera exigua. PLoS ONE 12(8): e0183054. (2017). https://doi.org/10.1371/journal.pone.0183054

Though most of our published work on EAB includes a dsGFP control, we agree that it could have been included here as a negative control to show that mortality was directly related to reduction of the target transcripts.  Future studies will use both dsGFP and HT115 cells as controls.

  1. The selection of the reference gene TEF should be described in the methods. While this gene scored well using the 4 algorithms, the data in the Supplemental data section do not provide details on the number of replicates, and they do not also show the Cq values for the two genes of interest (shi and hsp). The latter would have been informative to see how closely the target genes’ transcript levels matched those of the references. Why was only one reference gene selected, rather than use the geometric mean of more than one? This latter option is becoming the standard approach for qRT-PCR.

Most of the previously published emerald ash borer molecular work used TEF1α as the sole reference gene. Rajarapu et al. (2012) investigated the stability of reference genes in different EAB larval tissues, in different developmental stages, and under different experimental conditions, and identified TEF1α as the most stable gene to employ as an endogenous control to normalize mRNA levels in EAB. Since then, studies evaluating core RNAi machinery in EAB (Zhao et al. 2015) and EAB susceptibility to RNAi (Rodrigues et al. 2017, Rodrigues et al. 2018) utilized only TEF1α as the reference gene to normalize gene expression across samples. There are numerous instances in recent publications that use only a single reference gene in gene expression assays, including:

  • Zhang Yiqiu, Xu Letian, Li Shengchun and Zhang Jiang. Bacteria-Mediated RNA Interference for Management of Plagiodera versicolora. Insects. 10(12), 415; (2019) https://doi.org/10.3390/insects10120415

  • Dhandapani RK, Gurusamy D, Duan JJ, Palli SR. RNAi for management of Asian long‑horned beetle, Anoplophora glabripennis: identification of target genes. Journal of Pest Science. doi.org/10.1007/s10340-020-01197-8 (2020)

  • Majidiani S, PourAbad RF, Laudani F, et al.  RNAi in Tuta absolutamanagement: Effects of injection and root delivery of dsRNAs. Journal of Pest Science 92, 1409–1419. https://doi.org/10.1007/s10340-019-01097-6 (2019)

  • Haller S, Widmer F, Siegfried B D, Zhuo X and Romeis J. Responses of two ladybird beetle species (Coleoptera: Coccinellidae) to dietary RNAi. Pest Management Science 75(10): 2652-2662 (2019)

We have further tested the stability of reference genes TEF1α, ACT, β-TUB and GAPDH in experimental conditions fed with dsRNA and found TEF1α to be the most stable. That said, we agree that the MIQE guidelines should be followed in future work, but based on our findings and on recently published literature we believe that the use of TEF1α as the sole reference gene does not compromise our results.

  1. Re: Figure 2 c – there are many bands in the dsRNA lane 3 – what are the other bands and can they be ignored because those same bands are in lanes 1 and 2, which might be non-specific bands from bacteria that have failed to produce dsRNA? (please explain those two lanes).

The figure caption in the revision now contains a more complete description of the gels (Figs. 2b-d), which addresses this comment and those of Reviewer 2 (items 12 and 13).

“. . . lanes 1 and 2: uninduced dsSHI, lane 3: dsSHI induced by IPTG. . .” (rev. line 142)

  1. How were the E. coli culturing conditions optimized? Is there is certain time point after culturing to add the IPTG, and is there a certain time point to harvest bacteria to ensure maximal dsRNA production?

Optimization has been incorporated into section 2.3, bacterial transformation, in the manuscript revision:

Expression of T7 RNA polymerase was induced by adding a final concentration of 0.5mM of isopropyl-β-D-1-thiogalactopyranoside (IPTG). Bacteria with dsSHI and dsHSP were then incubated at 37°C, 30°C, and 25°C for up to 4h to evaluate dsRNA expression. Based on our optimization further experiments conducted at 37°C for dsSHI and 30°C for dsHSP. (rev. line 88-90).

In the original manuscript submission we also state that further optimization will be needed to move this approach forward to deployment (rev. line 221-222).

  1. QRT-PCR could be used to quantify dsRNA production, to provide a clear idea of the dose of dsRNA administered to the insects. As it stands, it is unclear what dose of dsRNA the insects received. Could the differences in mortality with the two dsRNAs (shi and hsp) be due to different dsRNA doses?

In our experiment 1ml of bacterial culture was centrifuged at 3000rpm for 15min and the pellet was dissolved in 100μl of 1% sucrose solution with green tracking dye (Kroger Co., USA). For each assay, 3μl of bacterial suspension were fed to individual neonate larvae using the droplet assay. Our initial study showed bacterially produced dsRNA, i.e. dsHSP and dsSHI, can cause significant mortality and reduction in target gene expression. Further studies need to be done to optimize the concentration of dsRNA administered orally.

However, our methods are consistent with Zhang et al. (2019), who used 50 µL of bacterial suspensions for bioassays and painted bacterial cells on fresh willow leaves and fed to first instar larvae of Plagiodera versicolor. Dhandapani et al. (2020) also delivered dsRNA expressing bacteria along with 10% sucrose solution to ALB adults and larvae. The methodologies in these studies are comparable to ours.

  • Zhang Yiqiu, Xu Letian, Li Shengchun and Zhang Jiang. Bacteria-Mediated RNA Interference for Management of Plagiodera versicolora. Insects. 10(12), 415; (2019) https://doi.org/10.3390/insects10120415

  • Dhandapani RK, Gurusamy D, Duan JJ, Palli SR. RNAi for management of Asian long‑horned beetle, Anoplophora glabripennis: identification of target genes. Journal of Pest Science. doi.org/10.1007/s10340-020-01197-8 (2020)

  1. Figure 3 is unnecessary – the same information is displayed in Figure 4.

Figure 3 shows final mortality on day 7, whereas Figure 4 shows survival over time.

  1. Why did the mortality assay end at day 7? It would have been ideal to see the number of treated larvae reaching adulthood, and see long-term effects.

Our approach is typical, and well documented in the literature. In many recent publications, dsRNA expressing bacterial feeding bioassays were recorded up to 7 days, including in:

  • Zhang Yiqiu, Xu Letian, Li Shengchun and Zhang Jiang. Bacteria-Mediated RNA Interference for Management of Plagiodera versicolora. Insects. 10(12), 415; (2019) https://doi.org/10.3390/insects10120415

  • Ganbaatar, O., Cao, B., Zhang, Y. et al.Knockdown of Mythimna separata chitinase genes via bacterial expression and oral delivery of RNAi effectors. BMC Biotechnol 17, 9 (2017). https://doi.org/10.1186/s12896-017-0328-7

We agree it would be interesting and informative to learn the long-term effects. However, EAB is a xylophagous endophage, and unlikely to survive to adulthood fed only sucrose, as in our experimental conditions.

  1. Please add the sample size/ replicates to the caption in Figure 4 – it easier than looking for it in the text.

Done. Rev. line 159.

  1. For Figure 5, three pictures do not really provide any quantitative evidence - why not have one picture and then a graph showing some statistically significant data?

We only have phenotypic data of the surviving neonate larvae fed dsRNAs in our bioassay.

  1. The authors suggest in their concluding remarks that resistance to dsRNAs can be easily overcome by selecting new targets. Some recent studies of dsRNA resistance in the corn rootworm have been published, so please revise this claim, based on those findings.

Done. This point has been incorporated into the conclusion, which now reads:

Additionally, management of resistance in the pest population is essential. Western corn root worm, Diabrotica virgifera has shown field level resistance to DvSnf7 dsRNA [29], necessitating development and selection of new targets. This process is relatively simple, however, and involves screening for and switching to other appropriate dsRNAs, thereby managing for potential development of resistance. (rev. line 226-230)

  1. The authors suggested that root drenching could be used to apply dsRNA to the trees, but will roots readily take up bacteria?

We’ve demonstrated proof of concept, but have not evaluated delivery methods. See also our response to Reviewer 2, item 18. We’ve modified the paragraph in the revision to address these concerns:

We’ve demonstrated that, when ingested, bacteria transformed to produce EAB-specific dsRNA can silence target genes and kill neonate EAB, which creates additional potential for its use as a biopesticide. Recombinant bacteria with EAB-specific dsRNA could be sprayed on foliage to be ingested by feeding beetles or on ash stems to be ingested by newly hatched neonates. Naked dsRNAs applied topically or through root drenching can be assimilated into and moved through ash plant tissues [26] suggesting crude extract of bacterially expressed dsRNA may also be able to be translocated through the plant via soil drench, trunk sprays or injection [27]. Recombinant bacteria producing EAB-specific dsRNAs also make genetic transformation of ash trees with dsRNA expressing constructs a more plausible goal [28]. (rev. line 209-217)

Reviewer 2 Report

This manuscript by Leelesh and Rieske describes experiments designed to create bacterial strains expressing dsRNA specific for two essential genes in the EAB (shibire and heat shock protein 70). The paper presents tests of the application of these strains to induce RNAi in EAB neonates. The goal of the work is to develop such materials as bio-pesticides for EAB control.

The paper presents the successful demonstration of RNAi with the bacterial strains produced and these results are of interest in terms of establishing RNAi parameters in this insect. However, in a number of ways, the authors are over-reaching in their conclusions based on the results presented.  The main problems are the lack of a non-specific dsRNA control, and a number of instances of claims that this approach can lead to bio-pesticides given the disconnect between the life stage tested and the insect’s destructive stage (mentioned in various individual comments below).

Specific individual comments:

Abstract:

Line 14 - the word "production" implies dsRNA, which has not been introduced. This should be reworded. The same applies in the next sentence, in that dsRNA is meant when RNAi is written. RNAi is a process, not a bio-pesticide.

Lines 21-23 - this logic has the flaw that RNAi was shown here with neonates, which feed on cambial tissue, but the use of E. coli expressing dsRNA as a bio-pesticide implies deployment against adults, that feed on leaves, and adults will be far more resistant to RNAi than neonates.

Introduction:

Line 38 – delivery via transgenic plants is also orally.

Line 54 - how is such an insecticide delivered given the larvae feed under the bark? This refers back to lines 21-23.

Materials & Methods:

Line 61 - spectrophotometry reveals little about RNA quality. It can measure "purity", but quality is usually meant to infer the RNA is intact. For that you use a gel or Bioanalyzer.

Line – 71 (Table 1) - were the qPCR amplicons placed outside the dsRNA sequences to ensure no carryover of signal?

Lines 74-76 - restriction enzyme nomenclature for Xba I is incorrect.

Line 86 - explain why different temperatures were used.

Line 99 - there should also be a control with non-specific dsRNA like green fluorescent protein (GFP) gene as a control for nonspecific dsRNA effects. The only control included, untransformed bacteria, is only useful as a control for bacterial toxicity.

Line 111 - why was mortality measured after 5 and 7 days, but gene expression was measured after 3 and 5 days? Please make it clear the time frames for both assays relative to the 5 days of dsRNA feeding.

Results:

Lines 125-126 - on what basis are the authors suggesting that dsRNA specific to the shi and hsp genes was produced? They have not blotted and probed them, they have not even indicated the size of the cloned inserts or expected sizes of the products?

Line 134 - what is in lanes 1 and 2?

Line 137 - a band is visible in lane 2, not lane 1.

(both of these experiments would be best presented in comparison with an un-induced culture. Are these in the other unspecified lanes?)

Line 144 - are there more quantitative data than a visual assessment of 3 individuals?

Line 160 - what is meant by "untreated" controls? Are these the insects feeding on untransformed bacteria? That isn’t really "untreated".

Line 164 - the genes are not being "depleted", their expression is being repressed.

Discussion:

Line 198 - indicate the identity of the INT gene.

Lines 200-203 - these suggestions are speculations only. Would bacteria be taken up from a soil drench? Would bacteria be translocated through the xylem the same as a dsRNA molecule? These questions are not addressed by this work.

Conclusions:

Lines 207-209 - this strain and strategy was developed by others and was not established by these authors. As to whether the quantities of dsRNA produced were large is debatable, based on the results in Fig 2, and the fact that the unequivocal identities of the products was not established.

Lines 209-212 - it could be convenient and cost-effective, but whether it would also be effective is debatable, given only the adults will feed on sprayed foliage, and they are not the destructive phase of the life cycle.

Lines 216-220 - this is a gross oversimplification. There are other means of resistance development that would in fact impair RNAi control and these should also be recognized.

Author Response

Response to Reviewers for insects-832830

Author responses to reviewer comments below in blue type; author changes in manuscript revision is in red type.

Reviewer 2

This manuscript by Leelesh and Rieske describes experiments designed to create bacterial strains expressing dsRNA specific for two essential genes in the EAB (shibire and heat shock protein 70). The paper presents tests of the application of these strains to induce RNAi in EAB neonates. The goal of the work is to develop such materials as bio-pesticides for EAB control.

The paper presents the successful demonstration of RNAi with the bacterial strains produced and these results are of interest in terms of establishing RNAi parameters in this insect. However, in a number of ways, the authors are over-reaching in their conclusions based on the results presented.  The main problems are the lack of a non-specific dsRNA control, and a number of instances of claims that this approach can lead to bio-pesticides given the disconnect between the life stage tested and the insect’s destructive stage (mentioned in various individual comments below).

Specific individual comments:

Abstract:

  1. Line 14 - the word "production" implies dsRNA, which has not been introduced. This should be reworded. The same applies in the next sentence, in that dsRNA is meant when RNAi is written. RNAi is a process, not a bio-pesticide.

This has been corrected, Rev. line 14-15.

  1. Lines 21-23 - this logic has the flaw that RNAi was shown here with neonates, which feed on cambial tissue, but the use of E. coli expressing dsRNA as a bio-pesticide implies deployment against adults, that feed on leaves, and adults will be far more resistant to RNAi than neonates.

We agree, and this was designed as a proof-of-concept study. However, the use of dsRNAs as biopesticides has potential against both larvae and adults, and clearly further studies are required. We’ve tempered our statement in the ms. rev., which now reads:

Our results suggest that ingestion of transformed E. coli expressing dsRNAs can induce an RNAi response in EAB. (rev. line 21-22)

Introduction:

  1. Line 38 – delivery via transgenic plants is also orally.

This has been clarified (rev. line 38) to read:

“. . . including by injection, orally, and through absorption . . .”

  1. Line 54 - how is such an insecticide delivered given the larvae feed under the bark? This refers back to lines 21-23.

Although we can envision this technology being deployed as a bark spray targeting newly hatching neonates, as well as a foliar spray targeting adults, we grant that our phrasing may be premature regarding the development of this technology. We’ve rephrased this in the revision to read:

In this study we evaluated the insecticidal potential of dsRNA expressing bacteria delivered orally to neonate EAB larvae. dsRNA expressed in bacteria could provide dual benefits in terms of inexpensive production and efficient delivery.  (rev. line 50-52)

Materials & Methods:

  1. Line 61 - spectrophotometry reveals little about RNA quality. It can measure "purity", but quality is usually meant to infer the RNA is intact. For that you use a gel or Bioanalyzer.

We agree; this has been clarified, and rev. line 63-64 now reads:

RNA concentration and purity were evaluated using a Nano Drop 1000 (Thermo Fisher, USA).

  1. Line – 71 (Table 1) - were the qPCR amplicons placed outside the dsRNA sequences to ensure no carryover of signal?

Our lack of explanation for this was an oversight. This is now clarified in the rev. l. 121-124:

In order to eliminate undesirable amplification from input recombinant plasmids and/or dsRNAs, primers for qPCR were designed to detect target mRNAs by amplifying only sequences that lay outside of the insert interfering sequences.

  1. Lines 74-76 - restriction enzyme nomenclature for Xba I is incorrect.

Corrected, rev. line 76-78.

  1. Line 86 - explain why different temperatures were used.

Done. Rev. l. 88-90 now reads:

Bacteria with dsSHI and dsHSP were then incubated at 37°C, 30°C, and 25°C for up to 4h to evaluate dsRNA expression. Based on this optimization, further experiments were conducted at 37°C for dsSHI, and 30°C for dsHSP.

  1. Line 99 - there should also be a control with non-specific dsRNA like green fluorescent protein (GFP) gene as a control for nonspecific dsRNA effects. The only control included, untransformed bacteria, is only useful as a control for bacterial toxicity.

Please see response to Reviewer 1, item 3.

  1. Line 111 - why was mortality measured after 5 and 7 days, but gene expression was measured after 3 and 5 days? Please make it clear the time frames for both assays relative to the 5 days of dsRNA feeding.

Mortality was measured on day 5 (the last day of dsRNA feeding) and on day 7 (the final day of bioassays). Rev. l. 105-106.

To evaluate gene expression, initially we collected samples after feeding neonates for 24, 48, 72 and 120h; we found that at 72 and 120h there was reduction in gene transcripts. Based on our preliminary work, further experiments focused on samples collected at 72 to 120h.

The intervals for our gene expression analysis were already stated in the original ms. submission (line 115), so no changes were made. Editor?

Results:

  1. Lines 125-126 - on what basis are the authors suggesting that dsRNA specific to the shi and hsp genes was produced? They have not blotted and probed them, they have not even indicated the size of the cloned inserts or expected sizes of the products?

We’ve included the size of the cloned insert in the manuscript revision, which now reads:

Using colony PCR, we confirmed that 100% of the recombinant bacterial colonies tested contained the insert: dsSHI (483bp) and dsHSP (468bp) (Figure 2b), and the IPTG induced bacteria expressed dsRNA specific to EAB (shi (483bp) and hsp (468bp). (rev. l. 131-132)

  1. Line 134 - what is in lanes 1 and 2?

The figure caption in the revision now contains a more complete description of the gels (Figs. 2b-d), which addresses this comment and the next one.

“. . . lanes 1 and 2: uninduced dsSHI, lane 3: dsSHI induced by IPTG. . .” (rev. line 142)

  1. Line 137 - a band is visible in lane 2, not lane 1. (both of these experiments would be best presented in comparison with an un-induced culture. Are these in the other unspecified lanes?)

“. . . lane 1: uninduced dsHSP, lane 2: dsHSP induced by IPTG).” (rev. line 144-145)

  1. Line 144 - are there more quantitative data than a visual assessment of 3 individuals?

We only have phenotypic data of the neonate larvae surviving after the dsRNA bioassay (fed with dsSHI).

  1. Line 160 - what is meant by "untreated" controls? Are these the insects feeding on untransformed bacteria? That isn’t really "untreated".

Yes, these are untransformed bacteria, and this has been clarified in the manuscript, which now reads:

However, 120 h post-exposure there was a 74.14% reduction in the transcript level, which differed significantly from controls (untransformed bacteria).  (rev. line 167-168)

  1. Line 164 - the genes are not being "depleted", their expression is being repressed.

This is corrected in the revision, which now reads:

Quantitative RT-PCR analysis of transcript levels after RNAi-mediated repression of gene expression in EAB. (rev. line 173-174)

Discussion

  1. Line 198 - indicate the identity of the INT gene.

Done. The manuscript now reads:

Ingestion of bacteria producing dsRNAs, specifically double stranded integrin (dsINT), has also been shown to reduce growth of the lepidoptera, S. exigua [22]. (rev. line 203)

  1. Lines 200-203 - these suggestions are speculations only. Would bacteria be taken up from a soil drench? Would bacteria be translocated through the xylem the same as a dsRNA molecule? These questions are not addressed by this work.

The authors agree! One of the purposes of a paper’s Discussion section is speculation, which is what stimulates further scientific inquiry.

This is a proof of concept paper; we have not conducted experiments on soil drench or bark sprays or stem injections using dsRNA producing bacteria. What we have done is demonstrate that we can transform bacteria to produce our target dsRNA, which silences target genes and causes insect mortality. That said, we’ve rephrased this paragraph to read:

We’ve demonstrated that, when ingested, bacteria transformed to produce EAB-specific dsRNA can silence target genes and kill neonate EAB, which creates additional potential for its use as a biopesticide. Recombinant bacteria with EAB-specific dsRNA could be sprayed on foliage to be ingested by feeding beetles or on ash stems to be ingested by newly hatched neonates. Naked dsRNAs applied topically or through root drenching can be assimilated into and moved through ash plant tissues [26] suggesting crude extract of bacterially expressed dsRNA may also be able to be translocated through the plant via soil drench, trunk sprays or injection [27]. Recombinant bacteria producing EAB-specific dsRNAs also make genetic transformation of ash trees with dsRNA expressing constructs a more plausible goal [28]. (rev. line 209-217)

       See also response to Reviewer 1, item 13.

Conclusions:

  1. Lines 207-209 - this strain and strategy was developed by others and was not established by these authors. As to whether the quantities of dsRNA produced were large is debatable, based on the results in Fig 2, and the fact that the unequivocal identities of the products was not established.

Yes, we agree, neither the bacterial strain nor the dsRNA delivery strategy was developed by us; this is not our claim. There are numerous publications, many of which we’ve cited, that utilize the strain and the strategy. What we have done is to demonstrate its application to a novel system that is in dire need of innovative management strategies, namely the EAB invasion of North America and the catastrophic tree loss associated with it. In our original submission we concede that optimization is one necessary next step in the path to deployment. (rev. line 221-222).

  1. Lines 209-212 - it could be convenient and cost-effective, but whether it would also be effective is debatable, given only the adults will feed on sprayed foliage, and they are not the destructive phase of the life cycle.

See response to Reviewer 1, item 13, and this reviewer, item 18. The authors believe that the utility of this approach will go beyond foliar sprays targeting adults (which, if it induces mortality would still reduce breeding populations). Bark sprays covering eggs could also be efficacious against newly hatching larvae, reducing tree damage. When the technology catches up we can envision translocation through plant tissues. Other than the modifications previously mention (rev. lines 209-215), no change to ms. here.

  1. Lines 216-220 - this is a gross oversimplification. There are other means of resistance development that would in fact impair RNAi control and these should also be recognized.

Done. This paragraph has been re-worked to include:

There are clearly knowledge gaps that must be addressed before this technology can be deployed in the EAB-ash system. Potential off target effects in EAB, mutation of RNAi core machinery genes, mutation of target genes, and enhanced dsRNA degradation, not to mention potential effects on non-target organisms, must be more fully understood before deployment of bacterially expressed dsRNA in EAB management is practical. (rev. line 232-236)

Round 2

Reviewer 1 Report

The authors have adequately addressed my previous questions, with just one minor exception. The only thing not fully explained is the insect rearing conditions - how were the adult insects reared and eggs collected? Please add a comment in the Methods. [I don't need to see the final edit, if this is the only adjustment made].

Reviewer 2 Report

The authors have addressed most issues in their revisions, however, a couple of points are not resolved. First, and most importantly, the study still lacks an appropriate control. The HT115 control is good to control for bacterial toxicity, but it does not serve as a control for general dsRNA toxicity (line 99 in original ms). To provide this control, another non-specific dsRNA like that for the GFP gene is needed. Neither promising to provide such a control in future studies, nor an example of an inappropriately designed study being published elsewhere, addresses the need for such a control in enabling the claim that the dsRNA sequence employed specifically produced the observed mortality.

Regarding lines 207-209 in the original manuscript, the authors agree, neither the bacterial strain nor the dsRNA delivery strategy was developed by them, so these should be referenced. Were the plasmid and strain obtained from AddGene? In that case acknowledgement can be made as instructed by AddGene.

Author Response

Response to Reviewers for insects-832830 revision 2

Author responses to reviewer comments below in red type; author changes in manuscript revision is in blue type.

Reviewer 2

  1. The authors have addressed most issues in their revisions, however, a couple of points are not resolved. First, and most importantly, the study still lacks an appropriate control. The HT115 control is good to control for bacterial toxicity, but it does not serve as a control for general dsRNA toxicity (line 99 in original ms). To provide this control, another non-specific dsRNA like that for the GFP gene is needed. Neither promising to provide such a control in future studies, nor an example of an inappropriately designed study being published elsewhere, addresses the need for such a control in enabling the claim that the dsRNA sequence employed specifically produced the observed mortality.

The use of dsGFP as a control for RNAi work with EAB is well substantiated in the literature [5,15,18], as well as in many other systems. The authors agree that it could have been included in the current study to show that mortality was directly related to reduction of the target transcripts. However, the control that we did employ demonstrates that the neonate EAB mortality observed is due to the dsRNA expressed by our manipulated bacteria, and not due to un-manipulated bacteria expressing no dsRNAs. The approach of using dsRNA-expressing versus non-expressing bacteria using HT115 cells as a control is well established, including:

Al Baki, A., J. K. Jung, and Y. Kim.  Alteration of insulin signaling to control insect pest by using transformed bacteria expressing dsRNA. Pest Manag. Sci. 2020, 76: 1020–1030.

This reference has been cited in the manuscript revision (rev. l. 102).

We’ve also tempered our claim in the Conclusions (rev. l. 220-221), which now reads:

“We show that EAB fed with dsRNA-expressing bacteria results in down regulation of selected genes, demonstrating the potential for application of bacterially-expressed dsRNA for controlling EAB.”

Again, we understand the utility of using both controls, and future studies will use both dsGFP and HT115. We are not in a position to repeat these experiments. Editor?

  1. Regarding lines 207-209 in the original manuscript, the authors agree, neither the bacterial strain nor the dsRNA delivery strategy was developed by them, so these should be referenced. Were the plasmid and strain obtained from AddGene? In that case acknowledgement can be made as instructed by AddGene.

The source of both the HT115 and the plasmid have been referenced in the revision, and AddGene added to the Acknowledgments (rev. l. 76, 258 for the plasmid; rev. l. *** for the HT115).

Round 3

Reviewer 2 Report

The primary problem of a missing control remains. If the authors wish to publish this work without a dsRNA control, then more revisions should be made to accommodate its absence than the single statement tempered in the Conclusions of version 3. Without a dsRNA control, there is no evidence that the mortality observed is not due to a non-specific effect rather than the gene expression changes measured. Upon rereading the latest version, there are two additional passages that would need to be tempered to accommodate this fact; lines 20-21 and 387-390.

To be transparent about the point, a statement should be included somewhere to the effect that in the absence of a non-specific dsRNA control it isn’t possible to conclude that the mortality observed was the result of the changes in gene expression measured. It is better to acknowledge the possibility than to be potentially misleading, given that it is not actually demonstrated without the control.
